Network-based analysis of differentially expressed genes in cerebrospinal fluid (CSF) and blood reveals new candidate genes for multiple sclerosis

Safari-Alighiarloo Nahid 1
Rezaei-Tavirani Mostafa tavirany@yahoo.com 1
Taghizadeh Mohammad 2
Tabatabaei Seyyed Mohammad 3
Namaki Saeed 4
1 Proteomics Research Center, Department of Basic Science, Faculty of Paramedical Sciences, Shahid Beheshti University of Medical Sciences , Tehran , Iran
2 Bioinformatics Department, Institute of Biochemistry and Biophysics, Tehran University , Tehran , Iran
3 Medical Informatics Department, Faculty of Paramedical Sciences, Shahid Beheshti University of Medical Sciences , Tehran , Iran
4 Immunology Department, Faculty of Medical Sciences, Shahid Beheshti University of Medical Sciences , Tehran , Iran
Constantin Gabriela
Electronic publication date: 2016 Dec 22
Publication date: 2016
Volume: 4
Electronic Location ID: e2775
Received 2016 Jul 5; Accepted 2016 Nov 8
Copyright: ©2016 Safari-Alighiarloo et al.
Copyright year: 2016
Copyright holder: Safari-Alighiarloo et al.
License: This is an open access article distributed under the terms of the Creative Commons Attribution License, which permits unrestricted use, distribution, reproduction and adaptation in any medium and for any purpose provided that it is properly attributed. For attribution, the original author(s), title, publication source (PeerJ) and either DOI or URL of the article must be cited.
License URL: https://creativecommons.org/licenses/by/4.0/

Keywords: Protein–protein interaction network (PPIN), Transcriptome, Topology, Modularity, Clique analysis, Multiple sclerosis

Funding: The authors received no funding for this work.

==============================
Background

The involvement of multiple genes and missing heritability, which are dominant in complex diseases such as multiple sclerosis (MS), entail using network biology to better elucidate their molecular basis and genetic factors. We therefore aimed to integrate interactome (protein–protein interaction (PPI)) and transcriptomes data to construct and analyze PPI networks for MS disease.

Methods

Gene expression profiles in paired cerebrospinal fluid (CSF) and peripheral blood mononuclear cells (PBMCs) samples from MS patients, sampled in relapse or remission and controls, were analyzed. Differentially expressed genes which determined only in CSF (MS vs. control) and PBMCs (relapse vs. remission) separately integrated with PPI data to construct the Query-Query PPI (QQPPI) networks. The networks were further analyzed to investigate more central genes, functional modules and complexes involved in MS progression.

Results

The networks were analyzed and high centrality genes were identified. Exploration of functional modules and complexes showed that the majority of high centrality genes incorporated in biological pathways driving MS pathogenesis. Proteasome and spliceosome were also noticeable in enriched pathways in PBMCs (relapse vs. remission) which were identified by both modularity and clique analyses. Finally, STK4, RB1, CDKN1A, CDK1, RAC1, EZH2, SDCBP genes in CSF (MS vs. control) and CDC37, MAP3K3, MYC genes in PBMCs (relapse vs. remission) were identified as potential candidate genes for MS, which were the more central genes involved in biological pathways.

Discussion

This study showed that network-based analysis could explicate the complex interplay between biological processes underlying MS. Furthermore, an experimental validation of candidate genes can lead to identification of potential therapeutic targets.

Introduction

Multiple sclerosis (MS) is a complex disease affecting the central nervous system (CNS) in which genetic, environmental and immunological factors are considered as its etiology (Ebers, 2008; Svejgaard, 2008). Although MS shows both autoimmune and neurodegenerative features, the pathophysiological processes which may occur both within and outside of the CNS remain obscure and don’t have an uniform distribution within the MS population (Brynedal et al., 2010). To study such complex diseases, which involved noticeably missing heritability (Goris & Liston, 2012; Manolio et al., 2009), it is more efficient to describe perturbed processes and dysregulated pathways rather to identify individual genes (Kim, Wuchty & Przytycka, 2011).

Transcriptome analysis of the target organ, i.e., the central nervous system, should reflect an unbiased survey of expression profiles for genes with altered transcript levels in disease states. Since it is difficult to achieve CNS samples, cerebrospinal fluid (CSF) cells have been used in many studies as a surrogate for the target organ in CNS disorders (Brynedal et al., 2010). Furthermore, peripheral blood mononuclear cells (PBMCs) are being considered as an easily accessible and informative source of biological material in MS transcriptome studies (Achiron et al., 2004; Bomprezzi et al., 2003; Singh et al., 2007). In this line, it has been reported that the study linking peripheral and CSF immune responses were essential to understand the immunopathogenesis of MS (Christensen et al., 2012).

Since the expression level change of a gene in a transcriptomic profile may be a result of an expression change of another gene and may not be the direct cause of the cellular phenotype, additional information is required to put them in context (Wachi, Yoneda & Wu, 2005). Network-based analyses of protein–protein interaction (PPI) or interactome delineate the known associations among proteins in the context of biochemistry, signal transduction and biomolecular networks (Rezaei-Tavirani et al., 2016; Zali & Rezaei-Tavirani, 2014; Wu et al., 2009). The integration analysis of large scale gene expression and PPI data will place the differentially expressed genes in the biological context (Bapat et al., 2010; Li et al., 2012; Safari-Alighiarloo et al., 2014; Safari-Alighiarloo et al., 2016). Protein networks reflect the functional grouping of interacting up/down regulated genes. The roles of the subsets of these genes, therefore, may be resolved using the combined data (Wachi, Yoneda & Wu, 2005).

Recently, the topological analyses have been applied to PPI networks by tools or algorithms such as modularity and centrality analyses by which the biological significance of proteins has been determined (Huang et al., 2013; Lee et al., 2011). Graph centrality measures like degree, betweenness and closeness centrality are very useful in the identification of nodes that are functionally crucial in the network by ranking the elements of the network (Hindumathi et al., 2014). In the PPI network the nodes with high degree defined as hub proteins and the nodes with high betweenness defined as bottleneck proteins, which both play a pivotal role in networks (Yu et al., 2007). The interest gene sets which usually presented as gene modules, protein complexes or pathways have been analyzed in integrative databases such as the Database for Annotation, Visualization and Integrated Discovery (DAVID) (Huang, Sherman & Lempicki, 2008), the Kyoto Encyclopedia of Genes and Genomes (KEGG) and Reactome to identify the sets of biological processes and molecular pathways of genes.

This study integrated transcriptome-interactome data to construct PPI networks for MS using abnormally expressed genes in paired CSF and PBMCs samples. Topological analyses were performed to determine the significant network biomarkers. Underlying biological processes and pathways have been sought by modularity and clique analyses. Finally, potential disease markers were identified, which were high centrality genes significantly involved in functional modules or complexes.

Methods

Transcriptome data collection and processing

Gene expression profiles in both CSF cells and PBMCs were obtained from the ArrayExpress Database under the accession number of EMTAB- 69 based on the Human Genome 133 plus 2.0 arrays (Brynedal et al., 2010). Accordingly, this study consisted of 26 multiple sclerosis patients, of whom 12 and 14 patients were sampled during relapse and remission, respectively. The MS patients were selected from a large cohort of newly diagnosed MS patients, and none of the patients had ever received immunomodulatory drugs. Control population included 18 subjects with other neurological diseases to assess MS specific transcriptome. The microarray raw data were converted to gene expression values using the RMA algorithm by the affy package within R software (Gautier et al., 2004). After preprocessing, each expression profile containing 54, 675 probe sets that ones with less discriminative power were removed according to the measurement of overall variance by the varFilter function using the genefilter package from the Bioconductor project within R software (Gentleman et al., 2011). After the preprocessing, a total of 27,336 probe sets from each sample were used for further analysis. To identify differential expression of the selected probes, the limma package in R software was used to perform the moderated t-test (Smyth, 2005). Where a gene had more than one probe on the microarray, the average expression value of all the related probes was used to estimate expression level of the gene.

Interactome data

The human PPI network was gathered from four major IMEx (Orchard et al., 2007) public databases: IntAct (Kerrien et al., 2012), MINT (Ceol et al., 2010), DIP (Xenarios et al., 2002) and InnateDB (Lynn et al., 2008). Indeed, public PPI databases which only stored experimentally verified interactions used to eliminate possible spurious interactions and avoid misleading conclusions. Our recent study showed IMEx databases (especially IntAct and DIP databases) had a greater number of significant correlations for their proteins’ topological features than the all other paired comparisons between BIND, HPRD, MINT, IntAct and DIP databases (Safari-Alighiarloo, Taghizadeh & Rezaei-Tavirani, 2015).

QQPPI networks construction and topological analysis

To construct Query-Query PPI (QQPPI) networks, the differentially expressed genes in CSF (MS vs. control) and PBMCs (relapse vs. remission) were separately located on human PPI network. QQPPI networks only consistent of the query genes as the nodes and direct interactions among them. The subnetworks of QQPPI were constructed and visualized by Cytoscape software (Shannon et al., 2003). Centrality parameters of QQPPI networks were analyzed using the Cytoscape and CentiBin softwares (Junker, Koschützki & Schreiber, 2006). The following parameters were calculated to determine biologically significant nodes (Zhang, 2009). Degree: the number of links to a given node. The Betweenness centrality of node v is calculated as: (1) CBv= ∑s≠t≠v∈Vρstvρst,

where the number of all shortest paths between node s and t regarded as ρst, and the number of shortest paths which passing through a node v out of ρst regarded as ρst(v). Indeed, this formula represents the ratio of the number of shortest paths passing through node v to the number of all shortest paths between s and t. The current flow betweenness centrality of a node v is the average of the current flow over all source–target pairs. Closeness centrality is defined as the reciprocal of the total distance from a node v, to all other nodes. Therefore, high values of closeness should indicate that all other nodes are in proximity to node v. (2) CCv=1∑u∈vdisu,v.

The centroid value is the most complex node centrality index and is computed by focusing the calculus on couples of nodes (v, w). The centroid value of an individual node ‘v’ is calculated by considering the number of nodes that have minimum shortest path which are closer to ‘v’ than ‘w’. A node v with the highest centroid value is the node with the highest number of neighbors separated by the shortest path to v, (3) Ccenv=minfv,w,

where fv,w=yvw−ywv and yv(w) denotes the number of nodes that are closer to v than w. Eigen vector centrality assign the relative significance of all nodes in the network by weighting connections to highly important nodes more than connections to nodes of low importance. (4) λCIV=ACIV,

where CIV donates the Eigen vector and λ donates the Eigen value.

Hub and bottleneck nodes were extracted from the networks in two steps; (1) In the networks, nodes with degree greater than or equal to the sum of mean and twice the standard deviation (S.D.), i.e., mean + 2*S.D. of the degree distribution, were considered as hubs (Ray, Ruan & Zhang, 2008). (2) We defined bottlenecks as the proteins that were in the top 5% in terms of betweenness centrality.

Identification and annotation of functional modules

Network clustering was implemented by Clustering with overlap neighborhood expansion (ClusterONE) algorithm in order to identify the connected regions within the networks with possible overlap (Nepusz, Yu & Paccanaro, 2012). The modules were identified to have a minimum density of >0.05 and a degree of >5. A cluster with a p-value of <0.05 was determined to be a module. The functional meaning for identified modules was further explored, and they considered as candidate functional modules if their genes were significantly enriched in the biological process of Gene Ontology (GO) annotation or KEGG pathway.

Identification of complexes containing clique

CFinder software was applied to extract biologically meaningful protein complexes from the PPI networks (Adamcsek et al., 2006). CFinder (http://www.cfinder.org/) was downloaded and implemented locally. Cliques with 3 nodes and 4 nodes (3-cliques, 4-cliques) were identified in the QQPPI networks by this software. The cliques were searched against CORUM database (Ruepp et al., 2010) to find significant protein complexes. Then, all the proteins associated with a specific complex were identified using the in house algorithm. Complexes containing 3 or more query proteins, as a cut-off, were listed in this study.

Functional enrichment analysis

An enrichment analysis was performed using Functional Annotation Chart in DAVID bioinformatics. To determine functional modules, only the enriched GO terms and pathways with p-values < 0.05 were considered significant. Furthermore, Cytoscape Enrichment Map plugin was used to visualize significant terms enriched in entire networks by following parameters: p-value cut-off = 0.001, q-value cut-off = 0.05, overlap coefficient cut-off = 0.6 (Merico et al., 2010).

Results

Expression analysis

We used the Limma package to analyze gene expression profile, E-MTAB-69, for comparison of four transcriptomes in MS (CSF: MS vs. controls and relapse vs. remission, PBMCs: MS vs. controls and relapse vs. remission). There were 3,062 genes with FDR < 0.05 whose expression was different in the CSF of MS patients as compared to the controls, but none in the respective PBMCs comparison. The number of up and down regulated genes was 1,080 and 1,982, respectively. On the contrary, when MS patients in relapse to those in remissions were compared, 1,163 differential expression genes with FDR ≤ 0.1 were seen in PBMCs, but none in the CSF. The number of up and down regulated genes was 301 and 762, respectively. The full lists of annotated differentially expressed probe sets are shown in Table S1 for the MS vs. control comparison in CSF cells, and in Table S2 for the relapse versus remission in PBMCs cells.

Networks’ topological analysis

We used only direct interactions of differentially expressed genes to construct QQPPI networks. The CSF PPI network consisted of 1,440 nodes and 3,500 edges and PBMCs PPI network involved 483 nodes and 941 edges. Topological features were processed to characterize the biology network from the random network. The power law of node degree distribution is one of most important criteria (Maslov & Sneppen, 2002; Zhu, Gerstein & Snyder, 2007). The distribution of node degree approximately followed power law distributions, where P(k) is a distribution of node degree, k is a degree and λ is a degree exponent, with λ = 1.94 and λ = 2.08 for CSF and PBMCs networks, respectively, and Fig. 1 indicates that the QQPPI networks were scale-free. The hubs and bottlenecks were extracted from the QQPPI networks by the criterion described in the method section (Table 1). Besides, we calculated four others centrality measurements involving closeness centrality, centroid value, Eigen vector centrality and current flow betweenness centrality and identified more central genes in the networks. The list of all nodes and their centrality measurements are prepared in Tables S3 and S4 for CSF and PBMCs in which candidate markers have been highlighted. The graphical structure of CSF and PBMCs PPI networks containing 5% top central genes are represented in Fig. 2.

Figure 1 The degree distribution of nodes followed power law distribution.

(A) Degree distribution of differentially expressed genes in CSF QQPPI network. (B) PBMCs QQPPI network. The graph represents a decreasing trend of degree distribution with an increase in the number of links showing scale-free topology.

Table 1 Hub-bottleneck identification.

Cut-off determination for hubs & number of hubs and bottlenecks.

	Mean (M)	Standard Deviation (S.D)	Cut-off (M + 2*S.D)	Number of hubs	Number of bottlenecks	
CSF	4.86	7.3	19.4	56	72	
PBMCs	3.89	6.06	16.01	20	25	

Figure 2 QQPPI networks generation by mapping of differentially expression genes on PPI data.

(A) CSF QQPPI network. (B) PBMCs QQPPI network. Nodes with high centrality measures are shown by bigger size than others. Green and red nodes represent proteins encoded by up- and down-regulated genes, respectively. Graphical representation of nodes was implemented by “Spring Embedded” layout in Cystoscape.

Modularity analysis

The ClusterONE algorithm was selectively implemented on CSF and PBMCs networks to mine the functional modules, which may reveal a lot of hidden biological significant processes. We further performed GO and pathway analysis using DAVID tool to characterize functional modules; 14 and six functional modules were discovered for CSF and PBMCs (p-value < 0.05), respectively. In the case of CSF, enriched modules were relevant to the comparison of MS versus controls in which modules correlated remarkably with many immune-related pathways such as, cytokine–cytokine receptor interaction, chemokine signaling pathway, Toll-like receptor signaling pathway, T cell receptor signaling pathway and Hematopoietic cell lineage. Further to them, some modules were enriched for apoptosis, p53 signaling pathway, MAPK signaling pathway, Hedgehog signaling pathway and Fc gamma R-mediated phagocytosis. The other major enriched pathways in modules included focal adhesion, cell cycle, endocytosis, gap junction, tight junction, ECM-receptor interaction, regulation of actin cytoskeleton (Table 2).

Table 2 Modularity analysis.

The list of pathways enriched in modules for CSF (MS vs. controls).

Module ID	Pathway	p-value	
M1	hsa04062 : chemokine signaling pathway	7.7E–3	
hsa04060 : cytokine–cytokine receptor interaction	1.5E–2	
hsa04672 : intestinal immune network for IgA production	3.8E–2	
M2	hsa05010 : Alzheimer’s disease	3.0E–3	
hsa05014 : Amyotrophic Lateral Sclerosis (ALS)	3.1E–2	
hsa04720 : long-term potentiation	4.0E–2	
M4	hsa04640 : hematopoietic cell lineage	4.7E–6	
hsa04060 : cytokine–cytokine receptor interaction	1.4E–4	
hsa04210 : apoptosis	8.6E–4	
hsa05020 : prion diseases	2.1E–2	
hsa05332 : graft-versus-host disease	2.3E–2	
hsa04940 : type I diabetes mellitus	2.5E–2	
M7	hsa00590 : arachidonic acid metabolism	3.3E–2	
M9	hsa04620 : toll-like receptor signaling pathway	1.7E–5	
M13	hsa05217 : basal cell carcinoma	2.3E–3	
hsa04340 : hedgehog signaling pathway	2.3E–3	
M14	hsa04010 : MAPK signaling pathway	1.5E–2	
M15	hsa04110 : cell cycle	1.2E–2	
M20	hsa04115 : p53 signaling pathway	3.3E–10	
hsa04110 : cell cycle	2.5E–8	
hsa04914 : progesterone-mediated oocyte maturation	1.8E–3	
M22	hsa04810 : regulation of actin cytoskeleton	1.6E–2	
hsa04666 : Fc gamma R-mediated phagocytosis	2.4E–2	
hsa04530 : tight junction	4.4E–2	
M23	hsa04144 : endocytosis	1.5E–2	
hsa04540 : gap junction	2.7E–2	
M25	hsa04510 : focal adhesion	2.7E–2	
hsa04512 : ECM-receptor interaction	3.1E–2	

Table 3 Modularity analysis.

The list of pathways enriched in modules for PBMCs (relapse vs. remission).

Module ID	Pathway	p-value	
M1	hsa04612 : antigen processing and presentation	6.6E–8	
hsa05340 : primary immunodeficiency	2.7E–2	
hsa05332 : graft-versus-host disease	3.0E–2	
hsa02010 : ABC transporters	3.4E–2	
M7	hsa04115 : p53 signaling pathway	3.5E–3	
hsa04110 : cell cycle	1.2E–2	
M8	hsa04623 : cytosolic DNA-sensing pathway	2.5E–4	
hsa04622 : RIG-I-like receptor signaling pathway	5.2E–4	
hsa04620 : toll-like receptor signaling pathway	1.5E–3	
M9	hsa04120 : ubiquitin mediated proteolysis	2.3E–2	
M10	hsa03050 : proteasome	1.3E–9	
M11	hsa03040 : spliceosome	1.0E–3	
hsa04350 :TGF-beta signaling pathway	4.1E–2	

For PBMCs, enriched modules corresponded to the comparison of relapse versus remission in which the majority of enriched pathways contributed to immune-related pathways like antigen processing and presentation, primary immunodeficiency, RIG-I-like receptor signaling pathway, Toll-like receptor signaling pathway and cytosolic DNA-sensing pathway. TGF-beta signaling pathway and p53 signaling pathway were the two noticeable signaling pathways in modules. The last enriched pathways were spliceosome, proteasome, ubiquitin mediated proteolysis and cell cycle (Table 3).

Identification of cliques and complexes

CFinder software was implemented to identify several 3-cliques and 4-cliques in the QQPPI networks. The corresponding complexes were retrieved from the CORUM database and shown in Table 4. In the case of CSF (MS vs. controls), these complexes mediated various biological functions such as protein processing (proteolytic), proteasomal degradation, stress response, protein binding, protease activator (ID: 32, 192 and 193), DNA conformation modification, transcription repression, protein modification by acetylation, deacetylation (ID:58), DNA conformation modification, transcription repression and posttranscriptional control (ID:105, 995,996 and 974), mitotic cell cycle and protein modification (ID:310 and 313), chromosome segregation/division (ID:1464), cell junction (ID:1839), actin cytoskeleton organization and biogenesis (ID:3008), ribosome biogenesis (ID:3055), protein modification and cellular signaling (MAPKKK cascade (ID:5909 and 5886).

Table 4 Clique analysis.

The list of complexes enriched for CSF and PBMCs.

Gene symbol	Complex	
CSF (MS vs. controls)	
PSMA3, PSMB1, PSMB3, PSMB9, PSME1, PSMD7	Proteasome (ID:39, 192,193)	
GPS2, NCOR2, TBL1X	SMRT complex (ID:58)	
EED, EZH2, RBBP4	Polycomb repressive complex 2,3 (PRC 2,3) (ID:105, 996,995), EED-EZH2 complex (ID:974)	
CCNB1, CCNB2, CCND1, CDK1, CDKN1A	Cell cycle kinase complex CDC2 (ID:310)	
CCND1,CCND3, CDKN1A	Cell cycle kinase complex CDK5 (ID:313)	
CBX5, DSN1, ZWINT	Mis12 centromere complex (ID:1464)	
CTNNA1, CTNNB1, SDCBP	SDCBP-CTNNB1-CTNNA1-CDH1 complex (ID:1839)	
TUBA1A, TUBA1B, TUBA1C	60S APC containing complex (ID:3008)	
IGF2BP1, ILF2, NOLC1, RPLP2, RPS11, RPS16, SRPK1, TUBA1A, YBX1	Nop56p-associated pre-rRNA complex (ID:3055)	
MAP2K1, SFN, YWHAG	Ksr1 complex (Ksr1, Mek, 14-3-3, Mapk), EGF stimulated (ID:5909, 5886)	
PBMCs (relapse vs. remission)	
PSMA1, PSMA2, PSMA7, PSMB10, PSMB3, PSMD3, PSMD4	Proteasome (ID: 38, 39, 181, 191, 192, 193, 194)	
DHX15, PABPC1, PRPF19, SF3B3, SNRPB	Spliceosome (ID:351)	
ACTB, ANXA6, MYH9, SPTAN1	PA700-20S-PA28 complex (ID:437)	
HNRNPH1, HNRNPM, PABPC1, PRPF19, SF3B3, SNRPB	C complex spliceosome (ID:1181)	
CDC37, HSP90AB1, MAP3K3	Kinase maturation complex 1 (ID:5199)	
CDC37, HSP90AB1, IKBKE	TNF-alpha/NF-kappa B signaling complex 8 (ID: 5269)	
CASP8 FADD FAS	FAS-FADD-CASP8 complex (ID: 5473, 5860), FAS-FADD-CASP8-CASP10 complex (ID: 5859), Death induced signaling complex DISC (ID: 5799, 5800)	
ACTB, MYH9, SPTAN1	Emerin complex 1 (ID: 5604)	

For PBMCs (relapse vs. remission), the identified complexes involved in many biological processes like protein processing (proteolytic), proteasomal degradation, stress response, protein binding, protease activator (ID: 38, 39, 181, 191, 192, 193, 194), RNA processing and RNA binding (ID: 351 and ID: 1181), protein targeting, sorting and translocation, protein transport and homeostasis (ID:437), protein kinase (ID:5199), NIK-I-kappaB/NF-kappaB cascade and cytokine activity (ID: 5269), apoptosis (ID: 5473, 5860, 5859, 5799 and 5800).

Functional enrichment analysis of the networks

To gain a full view of the networks potential functions, the networks’ nodes were annotated using the Functional Annotation Chart in DAVID and visualized using the Enrichment Map plugin in Cytoscape. As shown in Fig. 3, each node represented one functional annotation term. Nodes with more enriched genes were larger. Edge width was indicated the extent of overlapping between these categories (overlap coefficient cut-off 0.6). In case of CSF (MS vs. control), the ten most enriched entries in Gene Ontology (GO) biological processes were GO:0009611∼Response to wounding, GO:0006955∼Immune response, GO:0007242∼Intracellular signaling, GO:0010604∼Positive regulation of macromolecule metabolic processes, GO:0007049∼Cell cycle, GO:0002684∼Positive regulation of immune system processes, GO:0042981∼Regulation of apoptosis, GO:0006954∼Inflammatory response, GO:0016044∼Membrane organization, GO:0008283∼Cell proliferation. In case of PBMCs (relapse vs. remission), ten most enriched terms included GO:0043068∼Positive regulation of programmed cell death, GO:0006974∼Response to DNA damage stimulus, GO:0043065∼Positive regulation of apoptosis, GO:0006917∼Induction of apoptosis, GO:0044265∼Cellular macromolecule metabolic processes, GO:0033554∼Cellular response to stress, GO:0022402∼Cell cycle processes, GO:0010033∼Response to organic substance, GO:0002684∼Positive regulation of immune system process, GO:0051249∼Regulation of lymphocyte activation.

Figure 3 Functional categories of the networks were visualized using the Enrichment map plugin of the Cytoscape.

Significant biological processes are represented by one node in (A) CSF QQPPI network. (B) PBMCs QQPPI network. Nodes’ sizes indicate the significance of the enrichment (p-value). Edges show gene overlap between nodes and thickness indicates the number of overlapping enriched genes.

Mining and identification disease markers in modules and complexes

We screened more central nodes in CSF and PBMCs to investigate detailed analysis about their association in functional modules and complexes. Interestingly it was found that eight genes (STK4, RB1, CDKN1A, CDK1, RAC1, ARRB2, ARRB1, FN1) were located in functional modules and 15 genes (RBBP4, ILF2, RPS16, PSMA3, EED, EZH2, CDK1, CDKN1A, CTNNB1, SDCBP, SRPK1, TUBA1A, YBX1, SFN, YWHAG) were associated in complexes in CSF-QQPPI network. In case of PBMCs-QQPPI network, four genes (CDK2, PSMA1, IKBKE and MYC) were located in functional modules and genes (PSMA1, PSMA2, PSMA7, PSMB3, PSMD3, HNRNPM, FAS, ACTB, CDC37, HSP90AB1, MAP3K3, IKBKE, CASP8) were associated in complexes. Besides, it was obvious from the mining of functional modules and complexes that more central genes incorporated in driving pathways of MS pathogenesis (Table 5, also see Tables 2, 3 and 4). The more central genes and their expression values in CSF and PBMCs networks are illustrated in Fig. 4 and candidate markers are represented in Fig. 5. Since these differentially expressed genes in microarray dataset corresponded to topologically significant nodes in PPI networks, which have functional importance because of their involvement in functional modules and complexes, they called as candidate disease markers in our study.

Table 5 Central genes.

The list of more central genes enriched in functional modules and complexes for CSF and PBMCs.

Module/complex ID	Gene symbol	
CSF(MS vs. controls)	
M13	SMAD1	
M14	STK4	
M15	RB1	
M20	CDKN1A, CDK1	
M22	RAC1	
M23	ARRB2, ARRB1	
M25	FN1	
ID:39, 192,193	PSMA3	
ID:105, 974,996,995	EED, EZH2, RBBP4	
ID:310,313	CDK1, CDKN1A	
ID:1839	CTNNB1, SDCBP	
ID:3008	TUBA1A	
ID:3055	SRPK1, YBX1, ILF2, RPS16	
ID:5909, 5886	SFN, YWHAG	
PBMCs (relapse vs. remission)	
M7	CDK2	
M8	IKBKE	
M10	PSMA1	
M11	MYC	
ID: 38, 39, 181, 191, 192, 193, 194	PSMA1, PSMA2, PSMA7, PSMB3, PSMD3	
ID:1181	HNRNPM	
ID:437,5604	ACTB	
ID:5199, 5269	CDC37, HSP90AB1, MAP3K3, IKBKE	
ID: 5473, 5860, 5859, 5799, 5800	CASP8,FAS	

Figure 4 Nodes with high centrality measures which involved in significant biological pathways and their expression values.

More central nodes in (A) CSF QQPPI network. (B) PBMCs QQPPI network.

Figure 5 Candidate markers involved in functional modules and complexes.

The functional enrichment of candidate markers in (A) CSF QQPPI network. (B) PBMCs QQPPI network. Modules and complexes illustrated by brown and blue dotted circles, respectively.

Discussion

Although myriad genetic studies investigate the MS pathogenesis, our understanding have remained incomplete about the exact mechanism and its genetics (Baranzini et al., 2009; Zhang et al., 2011). The analysis of network-based biological data provides prominent tool to decipher the genetic basis of complex diseases by unraveling genes and processes not recognizable by genetic association approaches (Sharma et al., 2013; Yu et al., 2013). Due to implication of both intrathecal and peripheral immune activation in MS pathogenesis (Brynedal et al., 2010; Christensen et al., 2012; Veroni et al., 2015), we constructed PPI networks of abnormally expressed genes in paired tissues (CSF and PBMCs) for MS by integrating interactome and transcriptomes data. Using the analysis of the networks, we not only recognized several underlying biological processes, we also identified some important candidate markers for MS.

By studying two different tissues, neither being the actually affected tissue in MS, largely different but potentially significant and noticeable results have been achieved. Since the CNS is embedded in CSF, it is presupposed that CSF would represent more of the processes occurring within the CNS (Brynedal et al., 2010). Besides, the implication of peripheral immune response in MS progression has been substantiated (Kebir et al., 2009). The simultaneously obtained CSF and PBMCs samples showed only differentially expressed genes in CSF, comparing MS patients versus controls, and PBMCs comparing relapse versus remission. Lack of differential expression in CSF (relapse vs. remission) implies that outside events of the CNS such as pathogens and other environmental triggers not influencing the cells of the CSF, may primarily affect MS bouts (Brynedal et al., 2010). To further explore underling biological processes in MS, functional modules and complexes were characterized in these different tissues. More central genes involved in immune response, apoptosis, cell cycle and cell adhesion pathways, which considered as main biological processes in MS pathogenesis. Furthermore, in PBMCs (relapse vs. remission), proteasome and spliceosome complexes were enriched by both modularity and clique analyses and in CSF (MS vs. control) only proteasome enriched by clique analysis. The proteasome has crucial role in cell cycle progression and immune response (Basler et al., 2015). The immunoproteasome is a cytokine-induced variant of the 20S proteasome, which involved in the pathogenesis of autoimmune diseases and in the modulation of T helper cell differentiation (Basler, Kirk & Groettrup, 2013). Inhibition of the immunoproteasome subunit LMP7 (β5i) in animal models for autoimmune diseases including MS protected against these diseases (Basler et al., 2014). Spliceosome as basic machinery splicing, only enriched in relapse phase of MS in PBMCs. There is evidence to delineate the important role of alternative splicing in autoimmunity (Evsyukova et al., 2010).

In this work, we also identified the candidate genes for MS employing of topological analysis on the constructed networks in CSF and PBMCs tissues. These markers were further analyzed through a literature survey to confirm their potential contributions in MS pathogenesis. The compendium annotations for some of the most relevant candidate markers in terms of their expression are followed. In case of CSF, the first candidate was serine/threonine kinase 4 (STK4), also known MST1, enriched in MAPK signaling pathway (M14), and it was upregulated. The study of Konstantin et al. showed that genetic deletion of Mst1 altered T cell function and protected against autoimmunity as deletion of Mst1 reduced the severity of experimental autoimmune encephalomyelitis (EAE). Their results indicated that Mst1 regarded as a critical regulator of adaptive immune responses, Th1/Th2-dependent cytokine production and as a potential therapeutic target for immune disorders (Salojin et al., 2014). The second marker, RB transcriptional corepressor 1 (RB1) participated in cell cycle pathway (M7) and was downregulated. This gene is a negative regulator of the cell cycle, and acts as a transcriptional regulator (Indovina et al., 2013). It has been reported that Rb-mediated gene expression repression of E2F2 (transcription factor 2), by acting to tether Rb to specific E2F promoter sites, was crucial in T cells, and mutation of E2F2 in mice resulted in enhanced T lymphocyte proliferation leading to the development of autoimmunity (Murga et al., 2001). The third marker, cyclin-dependent kinase inhibitor 1A (CDKN1A) also named p21, was enriched in p53 signaling pathway, cell cycle (M20), and it was downregulated. The evidence exists for the p21 as a cell-cycle inhibitor that suppressed autoimmunity (Trakala et al., 2009). Indeed, recent studies disclosed that p21 as a specific regulator of the homeostasis of memory/activated T lymphocytes (Arias et al., 2007). The fourth marker, cyclin-dependent kinase 1 (CDK1) was enriched in p53 signaling pathway, cell cycle and progesterone-mediated oocyte maturation pathway (M20), and it was upregulated. The study of Yoshida et al. (2013) implicated that cyclin-dependent kinases were important regulators and potential targets for modulation of T cell immunity and tolerance. In this line, their results showed that CDK (including CDK1) inhibitors prohibited Th17 differentiation and expedited iTreg (induced regulatory T cells) development, which induced improving of experimental autoimmune encephalomyelitis in mice. The fifth marker, rho family, small GTP binding protein Rac1 (RAC1) was enriched in regulation of actin cytoskeleton, Fc gamma R-mediated phagocytosis (M22), and it was downregulated. Rac1 as a major player of the Rho family of small GTPases, have a key regulatory role in both actin and microtubule cytoskeletal dynamics, which it is central to axonal growth and stability, as well as dendrite and spine structural plasticity in the nervous system. Besides, it is also a substantive regulator of NADPH-dependent membrane oxidase (NOX) which is a main source of reactive oxygen species (ROS). Thereby, Rac1 plays a principle role in the inflammatory response and neurotoxicity mediated by microglia cells in the nervous system (D’Ambrosi et al., 2014). The study results displayed the relevance of Rac1 dysregulation in the pathogenesis of Amyotrophic Lateral Sclerosis (ALS) (D’Ambrosi et al., 2014). The sixth marker, enhancer of zeste homologue 2 (EZH2) involved in polycomb repressive complexes 2, 3 and EED-EZH2 complex, and it was upregulated. EZH2 catalyzes the addition of three methyl groups to lysine 27 of histone H3 (H3K27) in target gene promoters, which caused gene silencing. The Li & Jiang (2015) study showed that the activity of EZH2 must be consistently inhibited in neurons to evade re-entrance into a cell cycle process, and thus its overexpression could begin a pathway that ended in CNS neurodegeneration. The seventh marker, the syndecan binding protein (SDCBP) involved in SDCBP-CTNNB1-CTNNA1-CDH1 complex and the enriched biological process, was the cell junction; it was downregulated. SDCBP was shown to interact with syndecans, which aided cell adhesion and enhanced attraction and concentration of growth factors at the cell surface (Beekman & Coffer, 2008). Lopez-Ramirez et al. study represented that human brain endothelial permeability has been controlled by miR-155, which targeted molecules involved in cell-to-cell interactions such as SDCBP. They results indicated that miR-155_/_ mice showed lower levels of blood–brain barrier leakage in experimental autoimmune encephalomyelitis and an acute model of systemic inflammation (Lopez-Ramirez et al., 2014).

In case of PBMCs, the candidate markers corresponded to the comparison of relapse versus remission. The first candidate marker was cell division cycle 37 (CDC37). It was incorporated in complex TNF-alpha/NF-kappa B signaling complex 8 and downregulated. CDC37 and HSP90 is a member of IKK complex that disruption of the interaction between CDC37/HSP90 and IKK complexes impaired the activation of IKK and NF-κB in a TNF-dependent manner (Chen, Cao & Goeddel, 2002; Chen & Goeddel, 2002). NF-κB acts as a central mediator of immune and inflammatory responses, and it is involved in regulation of cell proliferation and apoptosis (Oeckinghaus & Ghosh, 2009). In this line, some studies disclosed that apoptosis was suppressed during acute relapse and this issue may lead to prolonged survival of autoreactive T cells (Achiron et al., 2007; Achiron et al., 2004). Downregulation of CDC37 could be remarkable issue during relapse stage with apoptosis suppression for further studies. The second marker, mitogen-activated protein kinase 3 (MAP3K3) involved in Kinase maturation complex 1, and it was upregulate. It is a member of MAPKs which implicated in all aspects of immune responses, from the initiation phase of innate immunity to activation of adaptive immunity (Dong, Davis & Flavell, 2002). The last marker, v-myc avian myelocytomatosis viral oncogene homolog (MYC) incorporated in TGF-beta signaling pathway (M11), and it was downregulated. Achiron et al. (2004) study showed that all components of the TGF-beta signaling pathway were underexpressed during MS pathogenesis. Since this pathway is known to prohibit cell proliferation and increase susceptibility to apoptosis induced by TGF-beta, their underexpression may be relevant to autoreactive T-cell expansion in MS patients.

Conclusions

This study showed the necessity of network-based analysis to get more insights in MS pathogenesis at post-genomic era. In summary, QQPPI networks of abnormally expressed genes in paired CSF (MS vs. control) and PBMCs (relapse vs. remission) samples were constructed for MS, and centrality, modularity and clique analyses have been implemented. Our results indicated that genes with high centrality in the networks incorporated into the main biological processes in MS progression at CSF and PBMCs. Furthermore, we identified several candidate genes via the systems biology viewpoint which might facilitate the identification of potential targets for the treatment of MS.

Supplemental Information

Supplemental Information 1 Supplementary file 1

Annotated differentially expressed probe sets in CSF

Click here for additional data file.

Supplemental Information 2 Supplementary file 2

Annotated differentially expressed probe sets in PBMCs

Click here for additional data file.

Supplemental Information 3 Six centrality measures for nodes in CSF QQPPI networks

Click here for additional data file.

Supplemental Information 4 Six centrality measures for nodes in PBMCs QQPPI network

Click here for additional data file.

Additional Information and Declarations

Competing Interests

Author Contributions

Data Availability

The authors declare there are no competing interests.

Nahid Safari-Alighiarloo conceived and designed the experiments, performed the experiments, analyzed the data, wrote the paper, prepared figures and/or tables.

Mostafa Rezaei-Tavirani and Mohammad Taghizadeh conceived and designed the experiments, contributed reagents/materials/analysis tools, reviewed drafts of the paper.

Seyyed Mohammad Tabatabaei performed the experiments.

Saeed Namaki conceived and designed the experiments.

The following information was supplied regarding data availability:

The raw data has been supplied as a Supplemental File.

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
