# Peer review of "Network-based analysis of differentially expressed genes in cerebrospinal fluid (CSF) and blood reveals new candidate genes for multiple sclerosis"

_PeerJ, doi:10.7717/peerj.2775_

## Round 0.1 · original submission · Major Revisions

The manuscript needs to be improved to provide more robust and novel data as indicated by review #1 and the biological message and novelty should be clearly underlined as requested by reviewer #2. Also, the treatments received by the MS patients and the references from the literature used to generate the networks need to be clearly stated.

·

Basic reporting

No comments

Experimental design

The analysis methodology is correctly described and applied. However, the overall bioinformatics approach should be improved as specified to the authors, to increase the rigorousness and novelty of the study.

Validity of the findings

The findings do suggest novel direction of investigation. However, further work is suggested to improve the validity of the study

Additional comments

In this study, Safari-Alighiarloo et al. performed a bioinformatics analysis on data obtained from the online database ArrayExpress. This is the first good feature of this work in which new evidence are sought, on data already published, applying a novel approach based on systems biology and in particular on network analysis. Approach like systems biology are very useful to investigate complex diseases, such us multiple sclerosis (MS), using a holistic approach instead of a more traditional one based on reductionism.
The authors focused on trascriptomics data obtained by the analysis of cerebrospinal fluid (CSF) cells, which represents the most accessible surrogate of central nervous systems (CNS) since it is difficult to achieve CNS samples and peripheral blood mononuclear cells (PBMCs), important in the immunopathogenesis of MS. The analyzed data provided different information, unfortunately not comparable, because the network obtained by CSF analysis compared healthy subjects vs MS patient instead of that of PBMCs in which patients in the relapsing vs remitting phase are analyzed.
From a methodological view point, they performed a correct and widely accepted network analysis procedure based upon network topological indexes. Particularly, they applied algorithms computing indexes of node centrality (degree and betweenness) and identifying topological clusters for the identification of most central nodes, functional modules and protein complexes respectively. Topological analysis was, correctly, coupled to data mining by interrogation of dedicated biological databases for protein-protein interaction and biological processes.

I have some observation about this study. The analysis is exclusively based on bioinformatics. No experimental validation is proposed. Thus, since this work is mainly focused on the topology analysis of the networks, in my opinion is too elementary and simplified. I suggest to use more complex centrality values in addition to “degree” and “betweenness” since in the last years, more parameters were developed. For example, “centroid value” could be better to describe the central position of a node instead of “degree” because this parameter provides a centrality index always weighted with the values of all other nodes in the graph. Basically, centroid does indicate the capability of a node to function as an “organizer” of interacting clusters (signalosomes), thus identifying a critical functional role for a protein. Indeed, the node with the highest “centroid value” is also the node with the highest number of reciprocally interacting neighbors (not only first, like degree) if compared with all other nodes.
Another useful centrality index is the “bridging” that includes also “weighted” betweenness. In biological terms, a protein with high bridging centrality value is functionally capable of holding together communicating proteins. Thus, a protein with high bridging centrality is a protein possibly bringing in communication sets of regulatory proteins.

The data obtained show that most of the differentially expressed genes, in paired CSF and PBMCs samples, participate to biological processes related to immune response, cell cycle, apoptosis and cell adhesion. In my opinion the results obtained give a good overview of the processes related to MS pathogenesis and confirm the potential of this kind of in silico analysis to get insight biological evidences. However, results are not a novelty. So I think that the enrichment analysis isn’t so informative and doesn’t’ add novel evidences about MS disease. Furthermore, modules and protein complexes identification, aiming at evaluating the implicated biological processes, is not necessary because in my experience also an enrichment analysis on the list of altered genes could give the same results. Moreover, it will be useful to extract clusters of nodes by using more than one methodology. I suggest the use of other algorithms like, for instance, MCODE or some of the algorithms contained in the ClusterMaker2 app.
My last advice is to try an enrichment on the entire network and create a figure of the enrichment result as a network, for example with EnrichmentMap, giving an additional information about the sharing of genes between terms.
To conclude, this work represents a good example of the potential of a systems biology approach based on network inference and analysis, but should be improved as suggested to provide more robust and novel data.

Reviewer 2 ·

Basic reporting

See below

Experimental design

See below

Validity of the findings

See below

Additional comments

This article reports on interesting results following a network-based analysis of differentially expressed genes in cerebrospinal fluid compared to blood in multiple sclerosis. The authors exploit already published data sets generated from patient samples and use a network-based analysis associated to the quantification of connectivity and betweenness of the pathology associated networks. They identify interesting functional modules as well as network differences between blood and CSF that may be relevant to the physiopathology of MS. However, I have several concerns about the manuscript at this stage.

1. In the methods, authors mention that none of the patients had received immuno modulatory drugs. How is this possible in a cohort of progressive MS patients? Authors need to give details in the manuscript in addition to citing the reference from the literature.

2. In the interactive dome data analysis methods, authors mention that the only used to experimentally verify the interactions: this needs to be explained since there is no experimental results in this manuscript.

3. In the results part of expression analysis, authors mention differentially expressed genes between PBMC and CSF. There is no clear message associated to this analysis. This needs to be specified. What is the interest and novelty of this analysis. Was this type of analysis already performed a in the published article by Brynedal et al? Is any novel finding reported?

4. In the following result section on topological analysis, the message once again is not clear. What is the conclusion of that part? What is the biological message?

5. In general, I don’t see any clear conclusion that would be biologically or clinically relevant. Authors identify differences by reconstructing networks based on transcriptomics data already generated by others. It is not surprising that different data sets from different tissues will be composed of genes that are organized in different types of structures and networks. The important is to analyzing dictates and provide the biological or clinical relevance to these differences which is currently not clear. Also the whole manuscript is based on reanalyzing one data set. It would be important to provide validation data set in order to make the conclusions more robust.

---

## Round 0.2 · accepted · Accept

The revised manuscript was significantly improved and the authors adequately addressed the criticisms raised by the reviewers.

·

Basic reporting

Overall, the analysis was improved with respect to the initial submission. The authors provided comprehensive feedback to my previous observations and suggestions.

Experimental design

The analysis did take advantage of new methods and algorithms to consolidate the findings.

Validity of the findings

Findings are now more compelling, even if the real originality of this study is in the procedure more than in the results, which, however, may have a sufficient impact.

Additional comments

The authors provided correct feedback to my previous observations and suggestions and provided convincing answer to all my concerns.